# Effect of Intensity Standardization on Deep Learning for WML Segmentation in Multi-Centre FLAIR MRI

Abdollah Ghazvanchahi[1]                                           AGHAZVAN@TORONTOMU.CA
Pejman Jahbedar Maralani[2]                          PEJMAN.MARALANI@SUNNYBROOK.CA
Alan R. Moody[2]                                          ALAN.MOODY@SUNNYBROOK.CA
April Khademi[1,3,4]                                            AKHADEMI@TORONTOMU.CA

[1] *Electrical, Computer and Biomedical Eng. Dept., Toronto Metropolitan University, Toronto, CAN*

[2] *Department of Medical Imaging, University of Toronto, Toronto, ON, CAN*

[3] *Keenan Research Center, St. Michael's Hospital, Toronto, ON, CAN*

[4] *Institute of Biomedical Engineering, Science and Technology (iBEST), Toronto, ON, CAN*

**Editors:** Accepted for publication at MIDL 2023

## Abstract

Deep learning (DL) methods for white matter lesion (WML) segmentation in MRI suffer a reduction in performance when applied on data from a scanner or center that is out-of-distribution (OOD) from the training data. This is critical for translation and widescale adoption, since current models cannot be readily applied to data from new institutions. In this work, we evaluate several intensity standardization methods for MRI as a preprocessing step for WML segmentation in multicentre Fluid-Attenuated Inversion Recovery (FLAIR) MRI. We evaluate a method specifically developed for FLAIR MRI called IAMLAB along with other popular normalization techniques such as Whitestrip, Nyul and Z-score. We proposed an Ensemble model that combines predictions from each of these models. A skip-connection UNet (SC UNet) was trained on the standardized images, as well as the original data and segmentation performance was evaluated over several dimensions. The training (in-distribution) data consists of a single study, of 60 volumes, and the test (OOD) data is 128 unseen volumes from three clinical cohorts. Results show IAMLAB and Ensemble provide higher WML segmentation performance compared to models from original data or other normalization methods. IAMLAB & Ensemble have the highest dice similarity coefficient (DSC) on the in-distribution data (0.78 & 0.80) and on clinical OOD data. DSC was significantly higher for IAMLAB compared to the original data ($p<0.05$) for all lesion categories (LL>25mL: 0.77 vs. 0.71; 10mL$\leq$ LL<25mL: 0.66 vs. 0.61; LL<10mL: 0.53 vs. 0.52). The IAMLAB and Ensemble normalization methods are mitigating MRI domain shift and are optimal for DL-based WML segmentation in unseen FLAIR data.

**Keywords:** Standardization, deep learning, WML, segmentation, FLAIR MRI

## 1. Introduction

White matter lesions (WML), or leukoaraiosis, are routinely found in the aging brain and are established cerebral vascular disease (CVD) markers (Wardlaw et al., 2015)(Pantoni, 2010)(Azizyan et al., 2011). WML represent increased and altered water content in hydrophobic white matter fibers and tracts. Changes in white matter vasculature likely contributes to WML pathogenesis (Gorelick et al., 2011). WML may be the result of ischemic injury from decreases in regional cerebral blood flow (Pantoni and Garcia, 1997). Demyelination and axonal degeneration have also been suggested as probable mechanisms (Wardlaw

et al., 2015). Typically, WML manifest as multifocal, diffuse periventricular or subcortical lesions of varying morphologies (Marek et al., 2018). The presence of WML is associated with cognitive decline, dementia, stroke, death, and lesion progression increases these risks (Debette and Markus, 2010)(Alber et al., 2019). Therefore, WML are significant clinical biomarkers for investigation.

In T2-weighted and fluid-attenuated inversion recovery (FLAIR) magnetic resonance images (MRI), WML appear as hyperintense signals in the cerebral white matter (Marek et al., 2018). FLAIR MRI is preferred for WML analysis (Azizyan et al., 2011), (Badji and Westman, 2020), (Wardlaw et al., 2013), since the high signal from the cerebrospinal fluid (CSF) in T2 is suppressed, thus highlighting white matter disease (Lao et al., 2008). This is due to increased water content secondary to ischemia and demyelination and much more robustly seen in FLAIR than with T1/T2 (Gorelick et al., 2011). WML classification is typically performed by a radiologist using visual rating systems such as the Fazekas scale (Fazekas et al., 1993) or by manual segmentation (Caligiuri et al., 2015). Manual segmentation is time-consuming, laborious, and has high inter and intra-variability (Caligiuri et al., 2015). For objective, consistent, and efficient WML analysis, automated WML segmentation methods have been the focus of extensive research efforts in recent decades.

There have been many WML frameworks in the past for FLAIR MRI, that consider unsupervised (Caligiuri et al., 2015) (Khademi et al., 2011)(Khademi et al., 2014), supervised (Anbeek et al., 2004) (De Boer et al., 2009) (Simões et al., 2013) (Knight et al., 2018) (Schmidt, 2017) and deep learning methods more recently. Comparisons of WML algorithms, such as in (Heinen et al., 2019), evaluated the performance of five automated WML segmentation methods in a multicentre FLAIR and T1 dataset. The methods mainly consisted of traditional machine learning (ML) algorithms and performance is reported for 60 volumes from six centres. Using similar WML segmentation methods, in (de Sitter et al., 2017), the authors investigate five WML segmentation tools for multiple sclerosis (MS) lesion segmentation using FLAIR and T1 images for 70 patients from six centres. In (Vanderbecq et al., 2020), the authors considered seven open source traditional WML segmentation methods for T1 and FLAIR and studied performance on research and clinical datasets. In (Frey et al., 2019), the authors provide a meta-review of the current WML segmentation methods applied in large-scale MRI studies.

One of the key limitations in machine learning models is poor testing performance on out of distribution (OOD) data - data that is not within the training distribution (In Distribution, ID). This is especially true for MRI, as variations in hardware and software create non-standard intensities, contrasts, and noise distributions across scanners. As shown in (Khademi et al., 2021), CNN algorithms typically perform the best for WML segmentation, but do not equally generalize across scanners and datasets. This domain gap is a significant problem for deployment and limits wide scale adoption, since models will not work equally well in new centres. One method to reduce the domain gap is intensity standardization (Reiche et al., 2019). Intensity standardization is the process of aligning the intensity histogram to some known distribution which maps the same tissues to the same intensity ranges. In this work, we evaluate several intensity normalization methods for FLAIR MRI, for WML segmentation performance on OOD data.

Table 1: FLAIR MRI ground truth datasets. All data is 3T and 3-5mm slice thickness.

| | Patient Information | | | | | |
|---|---|---|---|---|---|---|
| **Database** | **Disease** | **Volumes** | **Images** | **Patients** | **Centres** | **LL (mL)** |
| ADNI | Dementia | 35 | 1225 | 35 | 22 | $11.8 \pm 10.1$ |
| CAIN | Vascular | 63 | 3024 | 63 | 8 | $12.2 \pm 12.3$ |
| CCNA | Dementia | 30 | 1440 | 30 | 7 | $22.8 \pm 18.8$ |
| MICCAI | Vascular | 60 | 3580 | 60 | 3 | $17.6 \pm 17.4$ |
| Total | All | 188 | 9.27K | 188 | 39 | $15.0 \pm 15.2$ |
| | Acquisition Parameters | | | | | |
| **Database** | **GE/Phil./Siem.** | **TR (ms)** | **TE (ms)** | **TI (ms)** | **X (mm)** | **Y (mm)** |
| ADNI | 10/7/18 | 9000-11000 | 90-154 | 2250-2500 | 0.8594 | 0.8594 |
| CAIN | 12/35/16 | 9000-11000 | 117-150 | 2200-2800 | 0.4285-1 | 0.4285-1 |
| CCNA | 2/3/25 | 9000-9840 | 125-144 | 2250-2500 | 0.9375 | 0.9375 |
| MICCAI | 20/20/20 | 4800-11000 | 82-279 | 1650-2500 | 0.9583-1.2 | 0.9583-1.2 |
| Total | 44/65/79 | 4800-11000 | 82-279 | 1650-2800 | 0.4295-1.2 | 0.4295-1.2 |

## 2. Methods and Materials

### 2.1. Data

Experimental data for this work comes from 4 multicentre FLAIR MRI datasets for a total of 188 volumes with pixel-wise WML annotations. Sixty volumes from the MICCAI WML Segmentation Challenge (Kuijf et al., 2019) are used to train the models (ID) and the remaining is used for held-out OOD testing. The three OOD clinical datasets are from the Alzheimer's disease Neuroimaging Initiative (ADNI) (Aisen et al., 2015), the Canadian Atherosclerosis Imaging Network (CAIN) (Tardif et al., 2013), a pan-Canadian clinical study on vascular disease, and the Canadian Consortium on Neurodegeneration in Aging (CCNA), a pan-Canadian clinical study to analyze different types of dementia (Chertkow et al., 2019) (Mohaddes et al., 2018). Annotations for CAIN, ADNI and CCNA were developed by the authors. See (Khademi et al., 2021) for the annotation protocol and Figure 7 for inter-rater agreement between the two raters. Table 1 shows the acquisition parameters.

### 2.2. Intensity Standardization

Our original work in intensity standardization is performed to remove variability caused by the multicentre effect using a modified version of our original work in (Reiche et al., 2019) called IAMLAB. The original work performs 3×3 median filter denoising, bias field correction through lowpass filtering, and intensity standardization. Intensity standardization is achieved through a novel scaling factor that aligns the histogram modes of two volumes. As shown in (Reiche et al., 2019), the intensity intervals of tissues in 350K FLAIR MRI are more consistent across multicentre data using this approach. Slice refinement was removed which improves robustness since peak detection failed in upper and lower slices (and reduced alignment performance) and N4 bias field correction was used. Our method is compared to several other methods in the literature, including Nyul (Nyul and Udupa, 1999), which provides piece-wise histogram matching, z-score normalization and White Stripe (Shinohara et al., 2014), which provides a z-score normalization within a specific percentile.

## 2.3. WML Segmentation

The skip connection (SC) U-Net proposed in (Wu et al., 2019) is used in this work as it was found to be optimal for FLAIR-only WML segmentation (Khademi et al., 2021). SC UNet adds skip connections between the shallow and deep layers of a CNN architecture. The outputs from each max-pooling layer in the encoder arm are inputs for each transposed convolution layer in the decoder. Skip connections ease training through improved information and back-propagation flow (Wu et al., 2019), (Drozdzal et al., 2016) which has been shown to diminish vanishing gradients (Drozdzal et al., 2016). Generalized dice loss (Sudre et al., 2017), Adam Optimizer with a learning rate of 1e-4 over 100 epochs, and batch size of 64 were used. Images were patched into 64 x 64 regions with 50% overlap. Slight data augmentations were applied for rotation, scaling, shearing, scaling and translation (Li et al., 2018). Models were trained on a computer with a NVIDIA Tesla P100 GPU, 16GB RAM.

## 2.4. Performance Metrics

The KL-divergence is used to measure alignment between the average volume histogram of the dataset and each individual volume histogram. A low KL divergence indicates high alignment in intensities across the dataset. The evaluation metrics used in the MICCAI WML segmentation competition were used which includes the dice similarity coefficient (DSC), the H95, average volume difference (AVD), F1-score and Recall (Kuijf et al., 2019). The extra fraction (EF) was also used to measure the relative false-positive rate. To determine whether segmentation performance is significantly improved using intensity standardized data, a t-test is conducted between performance metrics for predictions from standardized and original data. Box-cox transformation is used to normalize distributions (except for AVD). Stochastic neighbour embedded (t-SNE) graphs (Hinton and Roweis, 2003) are also investigated to examine patterns in the data. The t-SNE method uses a pre-trained CNN and a projection of the feature representations onto two dimensions. Features similar to one another are overlapping in the feature space. The t-SNE graphs for original and normalized data are examined.

## 3. Results

The multicenter datasets listed in Table 1 are standardized using IAMLAB, white stripe, Nyul and z-score. SC U-Net was trained separately for original data as well as IAMLAB, Whitestripe, Nyul and Z-score standardized for WML segmentation, resulting in five models in total. An Ensemble method is considered, which takes pixel-wise majority vote across predictions generated by the five models trained on different intensity standardized images. The entire MICCAI dataset (which is balanced between GE, Philips and Siemens) is used for training all the models, and the held-out (unseen) clinical data (CAIN, CCNA, ADNI) are used to examine generalization. Three folds are used (approximately 67% for training, and 33% to testing) for all experiments. Prior to intensity standardization and WML segmentation, skull-stripping is performed on the volumes using U-Net for intracranial volume (ICV) segmentation (DiGregorio et al., 2021).

### 3.1. Intensity Standardization

Intensity standardized images are shown in Figure 9. Histograms of original and standardized volumes for WML only for all datasets and images are shown in Figure 1. To quantify the degree of alignment to the mean intensity distribution for each method, the KL-distance was computed and is shown in Figure 2. IAMLAB normalization has the best alignment (lowest KL) of all the methods, with KL = 0.06, compared to the original data with KL = 0.83. For reference, the intensity normalized histograms for the entire brain and FLAIR MRI slices of original and standardized image are shown in Figure 8 and 9. The t-SNE results for the various standardized and original datasets are shown in Figure 10, which shows features from different scanner vendors are more overlapping in the standardized images. The original data has non-overlapping clusters for the different scanners, indicating different feature mappings.

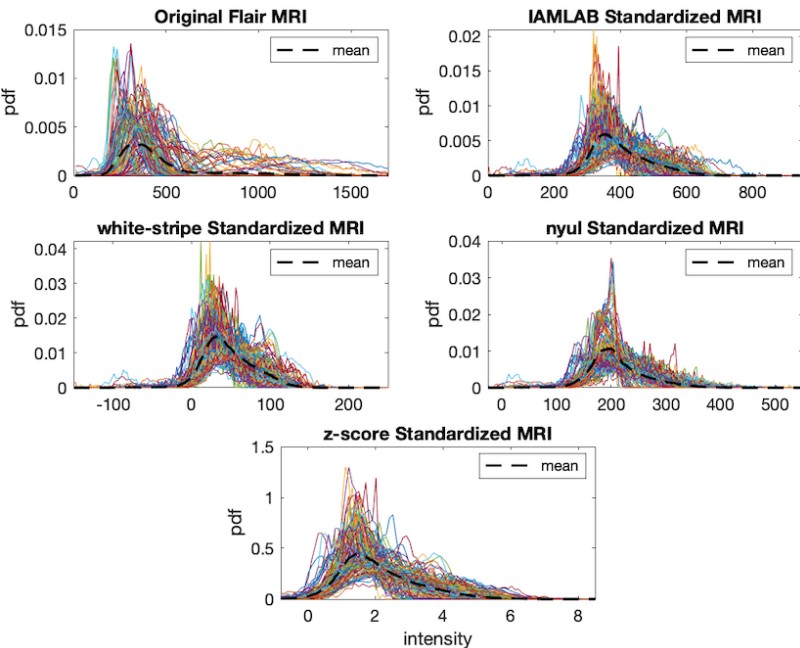

Figure 1: WML histograms of ground truth data over all normalization methods.

### 3.2. WML Segmentation: In Distribution

Model are verified using only the MICCAI dataset. MICCAI was divided into training/testing sets and evaluated over three fold cross validation and all 60 volumes were independently tested. Example segmentations are shown in Figure 3. As can be seen the model trained on original data misses some of the WML, which are detected on the normalized data. Mean validation metrics are shown in Table 2 where bold means best DSC. IAMLAB (0.78) and Ensemble (0.80) had the highest DSCs (original DSC = 0.75), which are similar to the top ranked teams in the MICCAI competition. The Ensemble method had leading metrics in other categories, except for EF, which original was lowest and likely

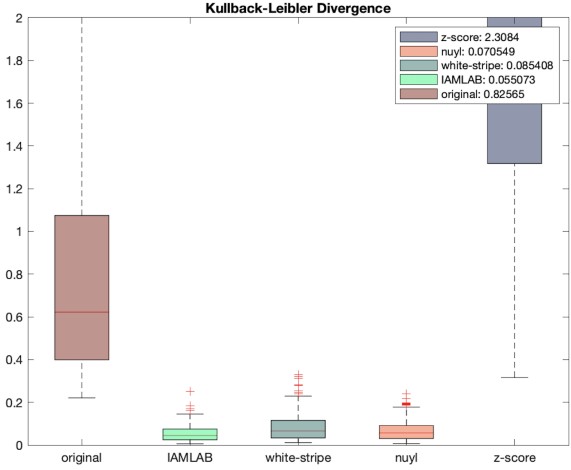

Figure 2: KL divergence for all metrics on the ground truth data.

related to under-estimation in Figure 3. Segmentation performance of top intensity standardized models (IAMLAB, Ensemble) were statistically different from the original data.

Table 2: Model validation: WML segmentation performance on 60 MICCAI. **Bold** is the best and * indicates mean of the metric is significantly different from the original data using t-tests (p<0.05).

|  | DSC | EF | H95 | AVD | F1-score | Recall |
|---|---|---|---|---|---|---|
| **Original** | 0.75 | **0.14** | 5.34 | 26.60 | 0.71 | 0.66 |
| **Nyul** | 0.76 | 0.16 | 5.32 | 22.47* | 0.68 | 0.63 |
| **White-Strip** | 0.78* | 0.17 | 5.25 | 19.72* | **0.72*** | 0.66 |
| **Z-Score** | 0.78* | 0.17 | 4.78 | 20.01* | 0.72* | 0.69 |
| **IAMLAB** | 0.78* | 0.18 | 4.59 | 20.08* | 0.71 | 0.69 |
| **Ensemble** | **0.80*** | 0.28 | **4.52*** | **19.46*** | 0.71 | **0.78*** |

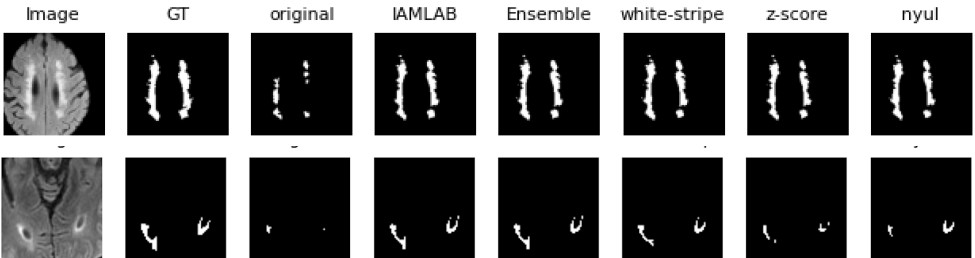

Figure 3: SC U-Net trained and tested on MICCAI (ID).

### 3.3. WML Segmentation: Out of Distribution

Next SC U-Net is trained using all 60 MICCAI volumes for the original and standardized data and tested on the held-out OOD clinical data from ADNI, CAIN and CCNA (128 volumes, approximately 5700 image slices). Example segmentations are shown in Figure 11 over all models and mean validation metrics are shown in Table 3. IAMLAB (0.64) and Ensemble (0.65) have the highest DSCs compared to the original (DSC = 0.60). Ensemble is also a top performer, with lowest EF (0.21) and H95 (11.21) and highest F1 score (0.60). Nyul has the highest Recall=0.76. When testing differences in DSC means, segmentation improvement afforded by intensity standardization was statistically different between original and all normalization methods, indicating segmentation improvement is significant.

To analyze the effect of IAMLAB and the Ensemble method on WML segmentation further, the change in DSC is plotted to investigate cases where segmentation was improved or hindered by normalization (Figure 4). The change in DSC is calculated by the DSC of the standardized model subtracted from the original data model for each volume. If there is a positive value, standardization improved performance while a negative value means it was more optimal to use original data and model. IAMLAB improved in 77% of the cases (98/128) and the Ensemble method improved in 86% of the cases. See example predictions in Figure 11, for cases with an average negative DSC change of -0.12 (A, B, C) and cases with an average positive DSC change of 0.17 (D-J) for IAMLAB standardized data compared to the original data. The improvement over most of the cases indicates standardization improves generalization to unseen data (OOD).

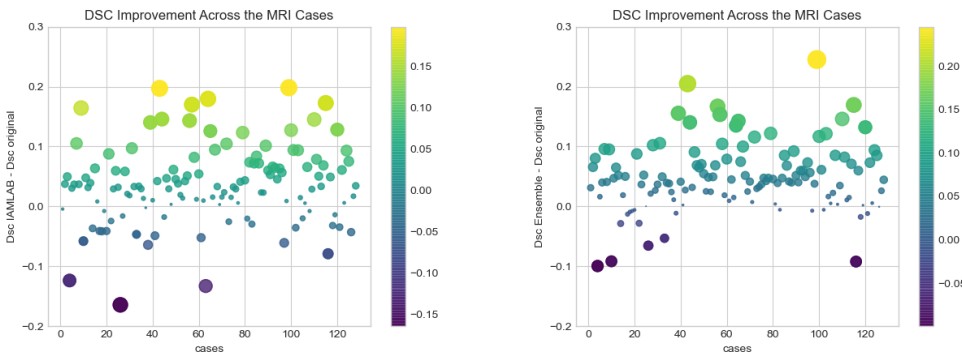

Figure 4: DSC Improvement in IAMLAB and Ensemble data compared to original

For WML segmentation problems, a small amount of improvement can be significant, since WML and the lesion loads can be small and missing a few pixels has a big impact on performance. Additionally, the small lesion load category is usually the most difficult for WML segmentation algorithms. To account for varying lesion loads, a secondary analysis was conducted over lesion load ranges as specified by the Fazekas scale (Van Straaten et al., 2006), which approximately corresponds to 0-10mL, 10-25mL and 25+mL for Fazekas 1, 2 and 3, respectively. Segmentation performance for these lesion load ranges is summarized in Table 4, 5 and 6. For LL <10mL, Ensemble had highest DSC (0.55) and IAMLAB had second

Table 3: WML segmentation performance on CCNA, CAIN, ADNI (N=128). **Bold** is the best and * indicates mean of the metric is significantly different from the original data using t-tests (p<0.05).

|  | DSC | EF | H95 | AVD | F1-score | Recall |
|---|---|---|---|---|---|---|
| **Original** | 0.60 | 0.21 | 13.44 | 35.16 | 0.56 | 0.69 |
| **Nyul** | 0.54 | 1.09 | 16.99 | 82.84 | 0.38 | **0.76\*** |
| **White-Stripe** | 0.62* | 0.48 | 15.06 | 33.89* | 0.53 | 0.73* |
| **Z-Score** | 0.62 | 0.55 | 13.61 | 44.19 | 0.48 | 0.73* |
| **IAMLAB** | **0.64\*** | 0.29 | 13.01 | **24.35\*** | 0.53* | 0.71* |
| **Ensemble** | **0.65\*** | **0.21** | **11.21\*** | 26.57* | **0.60\*** | 0.69 |

highest DSC (0.53). Compared to original data, DSC from IAMLAB and Ensemble were statistically different from the original data DSC, indicating the gains from standardization on OOD data are significant. For 10-25mL, Ensemble had the top DSC (0.67) and IAMLAB had the second highest DSC (0.66). For 25+mL, Ensemble had the highest DSC (0.77) by a large margin compared to the original data (0.71). For both groups, DSC was statistically different for IAMLAB and Ensemble compared to the original data. Models trained on the original data had lowest performance across the board especially in the large lesion group. Of all metrics, original data was best only in EF for 10-25mL and 25mL+ groups, which may be due to undersegmentation.

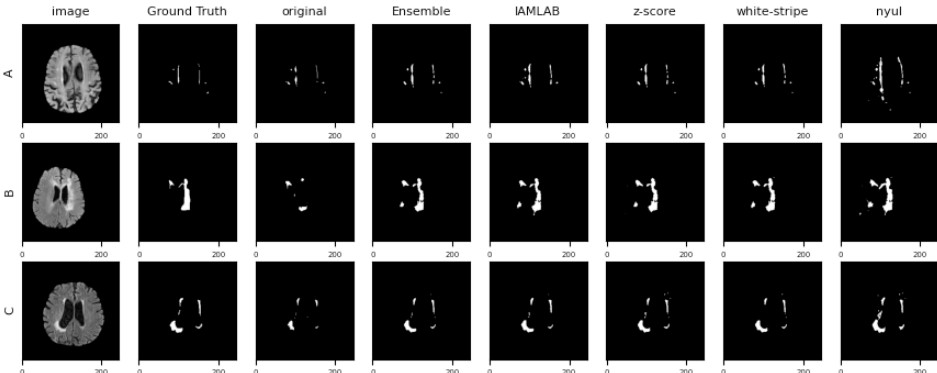

Figure 5: SC-UNET trained with MICCAI tested on ADNI (A), CAIN (B) and CCNA (C).

To investigate differences in normalization methods in terms of segmentation consistency, the coefficient of variation (CoV) of the DSC over different lesion loads are shown in Figure 6. Ensemble and IAMLAB methods have the highest consistency (lowest CoV) which indicates models developed on these datasets are more consistent and reliable. We postulate this is because these methods have better feature representation across imaging scanners due to aligned intensity profiles of the imaging volumes. This is supported by the t-SNE feature representations in the clinical datasets (CCNA, CAIN and ADNI), which are

unseen, OOD data, in Figure 10. Models trained on the original data have features that are more separated (and different) for each scanner type. In contrast, features extracted from the standardized data are more overlapping and similar across scanners likely leading to improved generalization in OOD data. This suggests intensity normalization minimizes the generalization gap between datasets for WML segmentation. It is interesting to note that Ensemble and IAMLAB consistently provide the highest DSC and lowest H95, AVD. These two algorithms may be providing complimentary information for optimal segmentation on OOD datasets. Overall, Ensemble and IAMLAB are the best performing algorithms on OOD for WML segmentation, which provides significant motivation for using intensity normalization methods for testing in unseen multicentre FLAIR MRI or when deploying algorithms on new scanners.

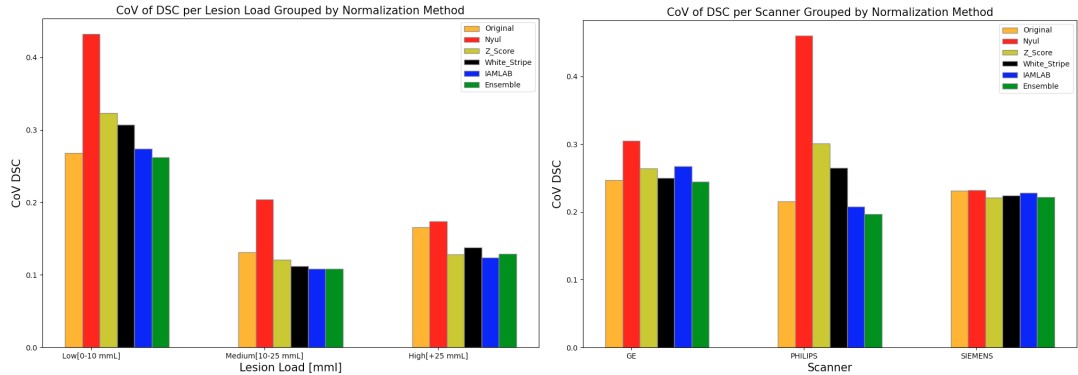

Figure 6: CoV of DSC metric from CAIN, CCNA and ADNI over lesion load (left) and scanner (right) group for different normalizations.

## 4. Conclusion

We investigate intensity normalization methods for deep learning-based WML segmentation methods on out-of-distribution FLAIR MRI datasets. An SC U-Net was trained using MIC-CAI competition data for the original dataset along with the four normalization methods. Models were tested on a diverse OOD test set from four different datasets comprising 128 imaging volumes. It was observed that intensity normalization using IAMLAB and the ensemble segmentation from IAMLAB, White-Stripe and Z-score normalization models leads to a statistically significant improvement in segmentation performance on OOD data. Therefore, IAMLAB and Ensemble methods are excellent candidates to improve generalization across clinical datasets from different centers, which is key for translation.

## Acknowledgments

We acknowledge the Natural Sciences and Engineering Research Council(NSERC) of Canada (Discovery Grant), Alzheimer's Society Research Program(ASRP) (New Investigator Grant) and the Ontario Government (Early Researcher Award) for funding this research.

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

# Appendix A. Graphs and Tables for Segmentation Result and Standardization

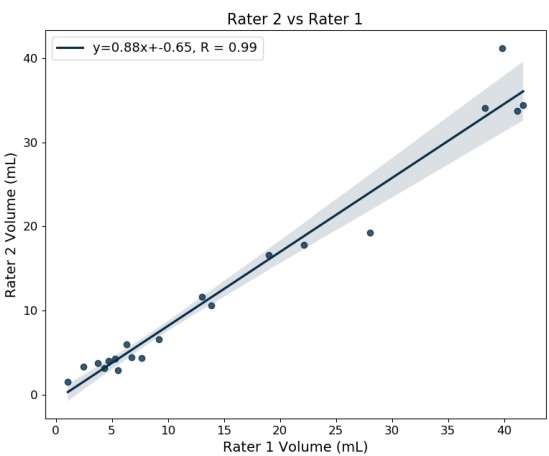

Figure 7: Inter-rater agreement between raters for WML annotations in CAIN, ADNI and CCNA.

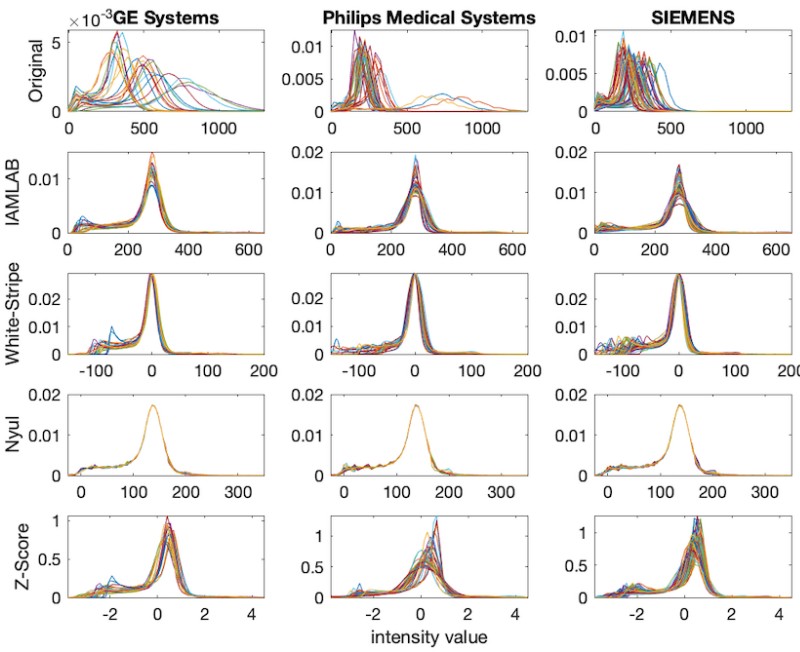

Figure 8: Whole Brain histograms of 128 volumes over all normalization methods.

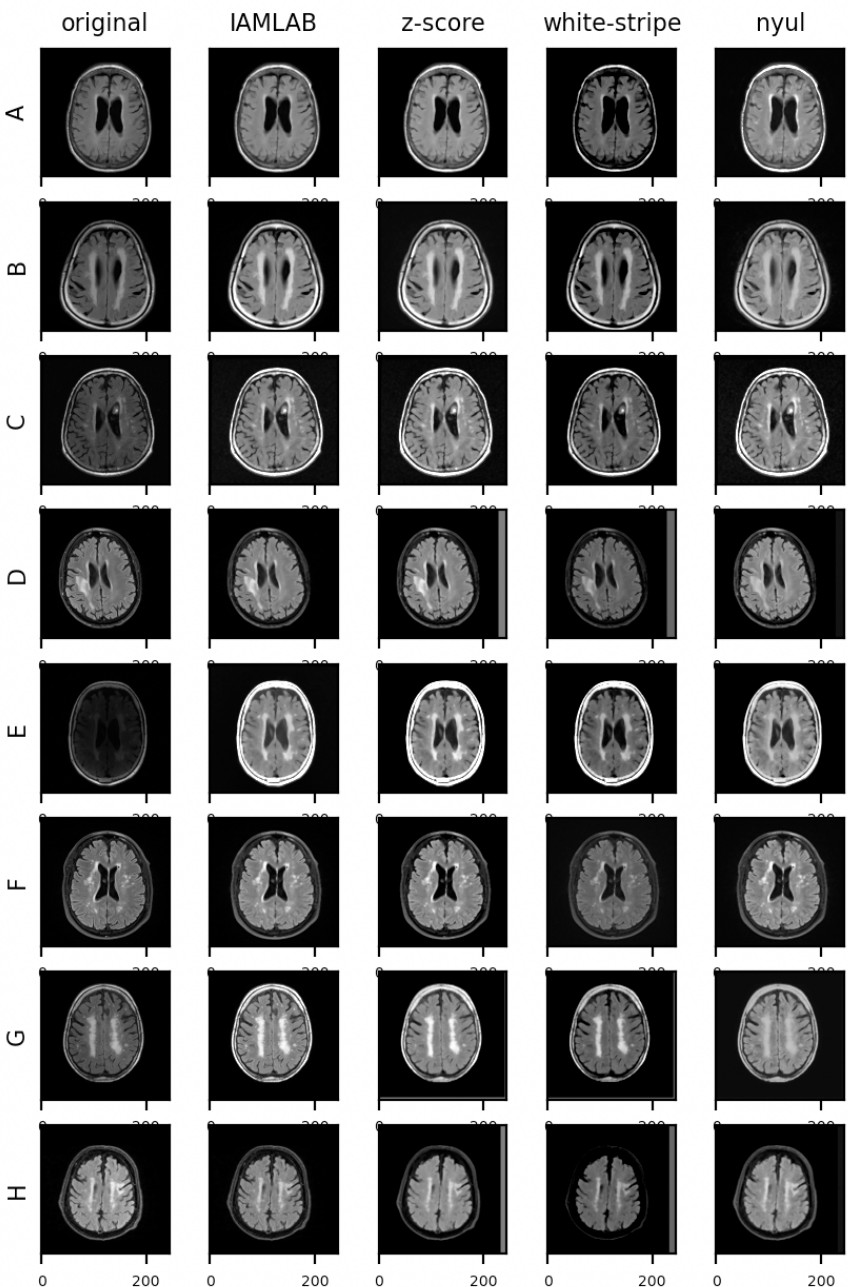

Figure 9: FLAIR MRI slices on 245x245 regions for original and all standardization. Images are from ADNI (A,B), CAIN (C,D) and CCNA (E,F) and MICCAI17 (G,H).

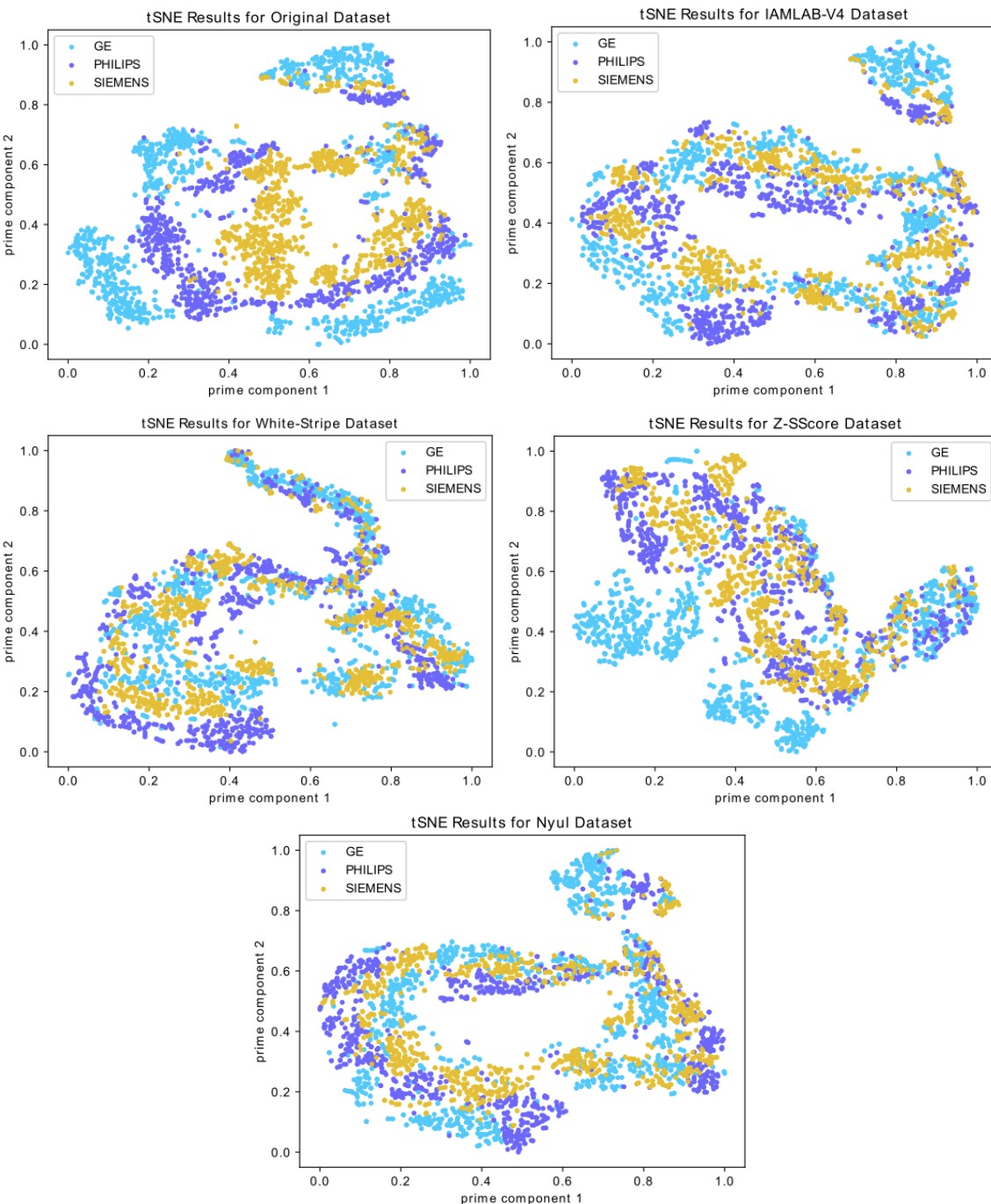

Figure 10: t-SNE representations for all data for original and standardized versions.

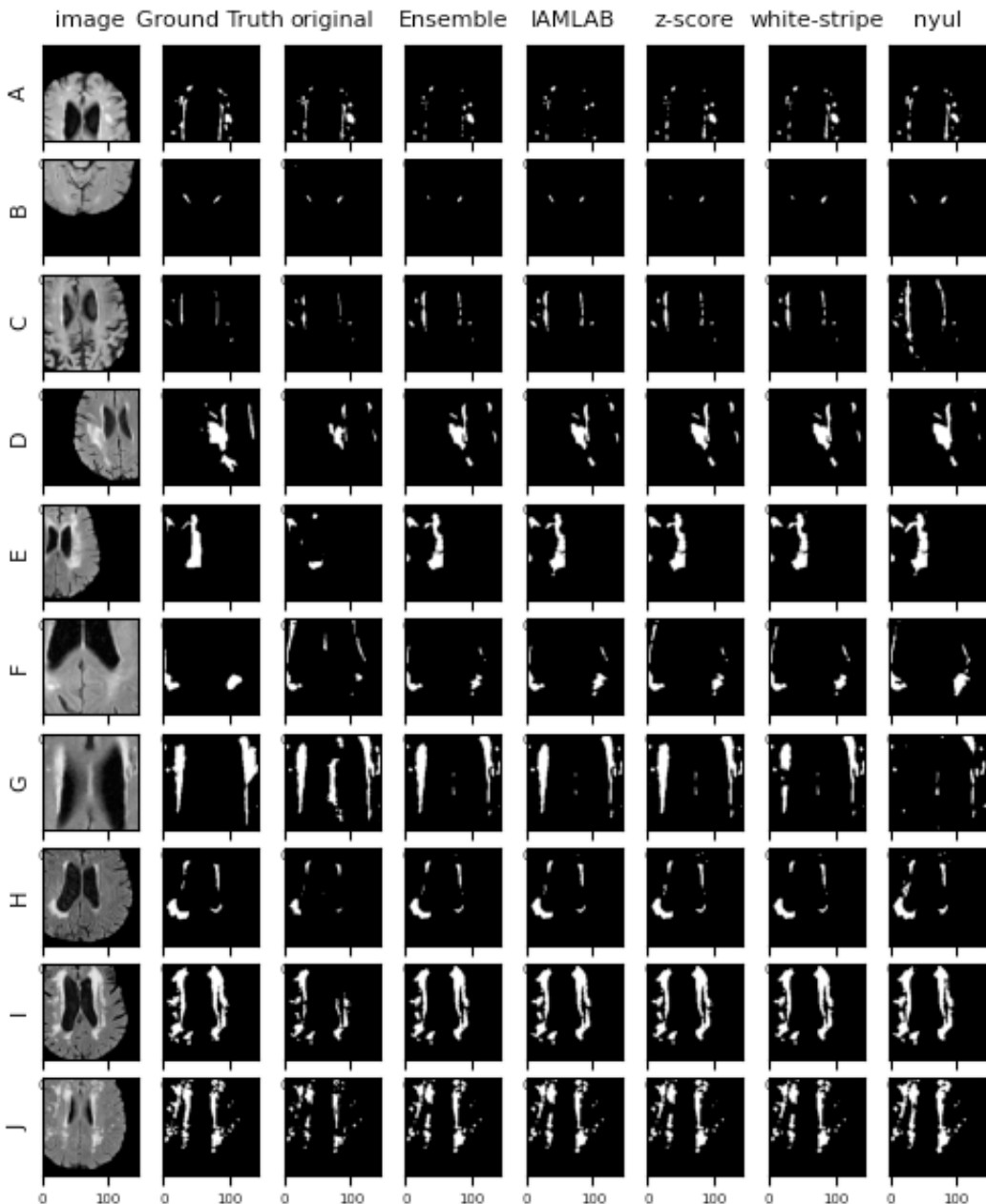

Figure 11: WML segmentation results on 145x145 regions for original and all normalizations. Images are from ADNI (A,B, C), CAIN (D,E,F,G) and CCNA (H,I,J).

Table 4: Segmentation performance on CCNA, CAIN, ADNI: LL<10mL (N=51)

|  | DSC | EF | H95 | AVD | F1-score | Recall |
|---|---|---|---|---|---|---|
| **Original** | 0.52 | 0.28 | 16.41 | 42.46 | 0.55 | 0.71 |
| **Nyul** | 0.41 | 1.90 | 23.76 | 156.98 | 0.28 | **0.79*** |
| **White-Stripe** | 0.52 | 0.73 | 21.65 | 54.96 | 0.48 | 0.74 |
| **Z-Score** | 0.50 | 0.88 | 19.10 | 79.48 | 0.44 | 0.74 |
| **IAMLAB** | 0.53* | 0.38 | 18.57 | **33.88** | 0.49 | 0.71 |
| **Ensemble** | **0.55*** | **0.25** | **15.90** | 36.02 | **0.57** | 0.69 |

Table 5: Segmentation performance on CCNA, CAIN, ADNI. LL: 10-25mL (N=45)

|  | DSC | EF | H95 | AVD | F1-score | Recall |
|---|---|---|---|---|---|---|
| **Original** | 0.62 | **0.18** | 12.62 | 34.20 | 0.59 | 0.70 |
| **Nyul** | 0.58 | 0.69 | 14.49 | 41.78 | 0.44 | **0.77*** |
| **White-Stripe** | 0.65 | 0.35 | 11.69* | 23.98* | 0.59 | 0.73* |
| **Z-Score** | 0.65 | 0.39 | 11.25* | 25.85* | 0.53 | 0.74 |
| **IAMLAB** | 0.66* | 0.26 | 10.26* | **19.63*** | 0.59* | 0.73* |
| **Ensemble** | **0.67*** | 0.20 | **9.01*** | 23.32* | **0.64** | 0.70 |

Table 6: Segmentation performance on CCNA, CAIN, ADNI. LL: 25mL+ (N=32)

|  | DSC | EF | H95 | AVD | F1-score | Recall |
|---|---|---|---|---|---|---|
| **Original** | 0.71 | **0.15** | 9.86 | 24.88 | 0.53 | 0.66 |
| **Nyul** | 0.71* | 0.37 | 9.74 | 22.43* | 0.44 | 0.69 |
| **White-Stripe** | 0.75* | 0.24 | 9.31 | 14.24* | 0.52 | 0.70 |
| **Z-Score** | 0.76* | 0.24 | 8.17* | **13.74*** | 0.48 | **0.70** |
| **IAMLAB** | 0.76* | 0.20 | 8.03* | 15.78* | 0.52 | 0.70 |
| **Ensemble** | **0.77*** | 0.15 | **6.82*** | 16.11* | **0.59*** | 0.68 |

