# OpenReview forum: "Effect of Intensity Standardization on Deep Learning for WML Segmentation in Multi-Centre FLAIR MRI"
_MIDL.io/2023/Conference — MIDL 2023 Poster_

### Official Review · Reviewer_yfHx · 2023-02-01

**Confidence:** 5
**Preliminary Rating:** 2

**Summary:**

The paper investigates the effects of intensity standardization techniques in the context of deep learning models of white-matter-lesion (WML) segmentation deployed on multicentre data. The paper evaluates five standardization strategies + an ensemble model using the MICCAI dataset as the source domain and a number of other datasets (ADNI, CAIN, CCNA) as the target domain in the experiments. The authors found that IAMLAB and ensemble gave the best results in their experiments.



**Strengths:**

The main strength of the paper, in my opinion, is that it is evaluated in a large number of datasets that is representative of clinical data. The authors also evaluate their results using a large number of representative image segmentation metrics.

The fact that IAMLAB seems to be a superior standardization technique is also interesting.

**Weaknesses:**

- Unfortunately, the authors identified themselves in the manuscript: "a modified version of our original work in (Reiche et al., 2019) called".

- Except for the technique proposed by the authors, the other methods are relatively old (1999, 2014). Aren't there more recent standardization techniques for comparison?

- In the motivation, the authors mention the high inter- and intra-rater variability, but they don't discuss how this variability impacts their analysis across different datasets. I imagine datasets were rated by different experts. Was it a single expert segmentation per image or a consensus panel?

- New standardization techniques are constantly being proposed. The paper would have a much higher impact if it had been proposed in the form of a benchmark. There is no mention of the availability of the code or the exact data split for reproducibility purposes.

- There are also more modern deep learning segmentation models than the skip-connection U-net. Would these results hold across different models?

**Deanonymize Review:**

no

**Detailed Comments:**

- The authors use a very large number of acronyms, some undefined, especially in the abstract;
- Color bar plots are not the best way to present the results in my opinion. They only give information about the mean values of the metrics. Why not use box plots which give more information to the reader?
- Overall, I thought the quality of the figures could be improved

**Paper Type:**

validation/application paper

**Questions To Address In The Rebuttal:**

I would ask the authors to address the questions and concerns I listed in the weaknesses section of my comments. I currently see the contribution of the paper as being limited. I would ask the authors to focus on inter- and intra-rater variability and reproducibility aspects of the work being presented.

---

### Official Review · Reviewer_rrbw · 2023-02-03

**Confidence:** 4
**Preliminary Rating:** 5
**Recommendation:** Oral

**Summary:**

The authors experiment with applying several intensity normalization techniques using a Skip Connection U-Net for segmenting White Matter Lesions in FLAIR MRI. Performance is evaluated on in-training distribution data, as well as on out-of-training distribution data. Results show that applying a combination of normalizations, namely IAMLAB and Ensemble, leads to improved segmentation performance both in in-training distribution and out-of-training distribution data.

**Strengths:**

The study adequately addresses the critical question of the performance of Deep Learning segmentation models in out-of-training distribution data. The authors report the segmentation performance of a U-Net variation on data from multiple centers by applying several normalization techniques. The data analysis is detailed, and the results fully support the discussion. Furthermore, the manuscript is well-written and reads at a high level.

**Weaknesses:**

The manuscript does not show any significant weaknesses. A slight rephrasing is needed for proper anonymization and references to the Appendix. Also, minor changes need to be made to Table 1 and Figure 1 to fix the formatting.

**Deanonymize Review:**

yes

**Detailed Comments:**

General
•	In the last paragraph of the Introduction and subsections 2.1 and 2.2, the text needs to be rephrased for proper anonymization.
•	The format of Table 1 looks weird like it is squeezed. Please adjust properly, as the rest of the Tables, to fix the formatting.
•	In Figure 1, the histograms on the left read hard. The subfigures are too small, while the scaling both on the x- and y- axes is different every time. Please either change the format (two rows with enlarged subfigures) or change the range of the axes to be the same (maybe not on the z-score, but all the others), or make a separate (larger) figure.
•	In Table 2, please add a sentence to explain what the asterisk indicates, as in Table 2.
•	Throughout the manuscript, please add that Figures 5-14 and Tables 4-6 are presented in the Appendix.

Abstract
•	Please introduce the abbreviation for SC-UNet, Skip Connection-UNet.
•	Please add a sentence to introduce the IAMLAB and Ensemble techniques.

Introduction
•	In paragraph three, the authors mention, “Comparisons of WML algorithms, such as in (Heinen et al., 2019), have emerged evaluating the performance of five automated WML segmentation methods in a multicentre FLAIR and T1 dataset”. It is unclear why this need has emerged and why it specifically emerged to evaluate approximately five methods. Please rephrase.
•	In the last sentence of the last paragraph, please change from “performance OOD data’ to “performance on OOD data”.
2.4. Performance Metrics
•	In the last sentence, the authors mention, “The t-SNE graphs for original and normalized data, across different scanners.” The sentence seems to be missing a reference to a Figure. Please add the reference.
3.3. WML Segmentation: In Distribution
•	The authors mention, “The Ensemble method had leading metrics in other categories, except for EF, which original was lowest and likely related to under-segmentation in Figure 2”. Please change to “The Ensemble method had leading metrics in other categories, except for EF, which originally was lowest and likely related to under-estimation in Figure 2.”


**Paper Type:**

validation/application paper

**Questions To Address In The Rebuttal:**

Evaluating the performance of Deep Learning segmentation models on out-of-training distribution data is addressed adequately by presenting several normalization techniques on several datasets. The authors have done extensive analysis to support their experiments. Therefore, the text needs minor revision upon acceptance without any further additions.

---

### Official Review · Reviewer_Hk7R · 2023-02-05

**Confidence:** 5
**Preliminary Rating:** 4
**Recommendation:** Poster

**Summary:**

The authors present an intensity normalization strategy to improve the segmentation model and compare various strategies to compare the results.
They trained the model on a public dataset including 60 scans and evaluated it on an independent cohort including 128 unseen volumes.
They claimed that the IAMLAB normalization method mitigates MRI domain shift and is optimal for WML segmentation in unseen FLAIR MRI datasets.


**Strengths:**

A comparison of multiple normalization methods for WMH segmentation.
This could be clinically relevant for improving the generalizability of WMH segmentation algorithms.
Extensive metrics to evaluate the model performance.

**Weaknesses:**

**The proposed normalization method is off-the-shelf, which is from the authors' previous work (MRI 2019)**.
This work is an additional validation of the previous method on the WMH segmentation task.

**The effectiveness of the ensemble model has been studied.**
In Table 2, there is no difference between the z-score and IAMLAB in in-domain data, while in Table 3, there is a difference. This is a very reasonable and nice result. But what is the rationale for showing IAMLAB+ensemble? Notably, the effectiveness of the ensemble model has been studied in [1].

**Only FLAIR is used for training and validation.** Notably, the original MICCAI-WMH challenge dataset contains FLAIR and T1. I understand that the authors use only FLAIR for external validation. But would such a normalization method work for both modalities?

References:
[1] Fully convolutional network ensembles for white matter hyperintensities segmentation in MR images. NeuroImage

**Deanonymize Review:**

yes

**Detailed Comments:**

**The proposed normalization method is off-the-shelf, which is from the authors' previous work (MRI 2019)**.
This work is an additional validation of the previous method on the WMH segmentation task.

**The effectiveness of the ensemble model has been studied.**
In Table 2, there is no difference between the z-score and IAMLAB in in-domain data, while in Table 3, there is a difference. This is a very reasonable and nice result. But what is the rationale for showing IAMLAB+ensemble? Notably, the effectiveness of the ensemble model has been studied in [1].

**Only FLAIR is used for training and validation.** Notably, the original MICCAI-WMH challenge dataset contains FLAIR and T1. I understand that the authors use only FLAIR for external validation. But would such a normalization method work for both modalities?

References:
[1] Fully convolutional network ensembles for white matter hyperintensities segmentation in MR images. NeuroImage

**Paper Type:**

validation/application paper

**Questions To Address In The Rebuttal:**

**This is not an anonymous submission**
In Section 2.2, the authors say, 'Intensity standardization is performed to remove variability caused by the multicentre effect using a modified version of our original work (Reiche et al., 2019) called IAMLAB'. This is against the double-blind rule during the review.

**The proposed normalization method is off-the-shelf, which is from the authors' previous work (MRI 2019)**.

**The effectiveness of the ensemble model has been studied.**

**Only FLAIR is used for training and validation**

---

### Meta-Review · Area_Chair_xNtC · 2023-02-23

**Recommendation:** Accept (Poster)
**Confidence:** 4

**Metareview:**

Based on the comments of the reviewers and my own reading of the paper, the paper evaluates multiple intensity standardization methods for FLAIR MR images in the context of segmenting white matter lesions in the brain. The main strength is that the evaluation in performed on multiple datasets, and the evaluation is quite comprehensive. On the weaknesses side, there is no novelty in this paper: it is essentially evaluation of multiple off-the-shelf methods (including a slightly modified version of their own previous work from 2019) to this particular problem. The authors also "proposed" an ensemble model, but it is trivially straightforward as well. Additionally, no code is publicly available (pending patent), so it is not clear if it will be of any use to the greater community in near future.